# Effects of polyploidization and their evolutionary implications are revealed by heritable polyploidy in the haplodiploid wasp *Nasonia vitripennis*

**Kelley Leung** [ORCID]*, **Louis van de Zande**⊕, **Leo W. Beukeboom**⊕

Groningen Institute for Evolutionary Life Sciences, University of Groningen, Groningen, The Netherlands

⊕ These authors contributed equally to this work.

* k.leung@rug.nl

**Data Availability Statement:** All relevant data are within the paper and its Supporting Information files.

**Funding:** This study was funded by the Marie Curie Innovative Training Network BINGO (Breeding

## Abstract

Recurrent polyploidization occurred in the evolutionary history of most Eukaryota. However, how neopolyploid detriment (sterility, gigantism, gene dosage imbalances) has been overcome and even been bridged to evolutionary advantage (gene network diversification, mass radiation, range expansion) is largely unknown, particularly for animals. We used the parasitoid wasp *Nasonia vitripennis*, a rare insect system with heritable polyploidy, to begin addressing this knowledge gap. In Hymenoptera the sexes have different ploidies (haploid males, diploid females) and neopolyploids (diploid males, triploid females) occur for various species. Although such polyploids are usually sterile, those of *N. vitripennis* are reproductively capable and can even establish stable polyploid lines. To assess the effects of polyploidization, we compared a long-established polyploid line, the Whiting polyploid line (WPL) and a newly generated *transformer* knockdown line (tKDL) for fitness traits, absolute gene expression, and cell size and number. WPL polyploids have high male fitness and low female fecundity, while tKDL polyploids have poor male mate competition ability and high fertility. WPL has larger cells and cell number reduction, but the tKDL does not differ in this respect. Expression analyses of two housekeeping genes indicated that gene dosage is linked to sex irrespective of ploidy. Our study suggests that polyploid phenotypic variation may explain why some polyploid lineages thrive and others die out; a commonly proposed but difficult-to-test hypothesis. This documentation of diploid males (tKDL) with impaired competitive mating ability; triploid females with high fitness variation; and hymenopteran sexual dosage compensation (despite the lack of sex chromosomes) all challenges general assumptions on hymenopteran biology. We conclude that polyploidization is dependent on the duplicated genome characteristics and that genomes of different lines are unequally suited to survive diploidization. These results demonstrate the utility of *N. vitripennis* for delineating mechanisms of animal polyploid evolution, analogous to more advanced polyploid plant models.

Invertebrates for Next Generation BioControl, project 641456; https://cordis.europa.eu/project/id/641456), awarded to coordinator BAP (https://www.wur.nl/nl/Personen/Bart-dr.ir.-BA-Bart-Pannebakker.htm) and project advisors LWB (https://www.rug.nl/staff/l.w.beukeboom/) and LvdZ (https://www.rug.nl/staff/louis.van.de.zande/). The funders had no role in study design, data collection and analysis, decision to publish, or preparation of the manuscript.

**Competing interests:** The authors have declared that no competing interests exist.

## Introduction

Polyploidization by whole genome duplication (WGD) causes numerous phenotypic irregularities including cell to whole body gigantism [1] disruption of epigenetic mechanisms (especially sexual dosage compensation) [2–4], and sterility due to meiotic error and aneuploidies [5]. For these reasons, polyploidy was once considered a rare and catastrophic cellular event with negligible contribution to evolution [2, 5–7]. However, in stark reversal, polyploidy is now recognized as a powerful evolutionary driver despite the significant likelihood of neopolyploid inviability. Investigation of gene duplicates [8] has led to the inference that most Eukaryotic groups have polyploid ancestry. Having extra gene copies apparently provisioned more advanced "evolutionary toolboxes," with novel gene copies undergoing neofunctionalization or subfunctionalization [2, 9, 10]. As lineages underwent "re-diploidization," diversified gene networks remained, conferring major evolutionary advantages [11, 12] such as resistance to environmental stress and corresponding range expansions [13, 14], as well as spurring mass speciation events [9, 15]. How the disadvantaged neopolyploid individual leads to a heavily advantaged evolutionary polyploid lineage has been a major theme in evolutionary research.

This question has been widely studied in plants. Up to 90 percent of all plant species are ancestral polyploids [16], including many crop species that have better yield and hardiness when newly polyploidized [17]. It has been suggested that plants are less subject to the detrimental effects of polyploidization than animals. For example, they lack both sex chromosomes and sexual dosage compensation mechanisms that can be disrupted with ploidy increases [2, 18]. They are also thought to be more plastic in their development and less affected by cellular and body plan changes [19]. The prominence of plant polyploidy over animal polyploidy has been to a degree debunked. Ancestral WGDs have been identified for numerous branches of Eukaryota, with two notably occurring at the base of the vertebrate branch [20–22]. Regardless, research on animal polyploid evolution has focused on certain groups with either prominently known or relatively frequent WGDs. These include the fish [23] and animals with reproductive strategies that reduce problems of offspring viability (e.g., parthenogenesis [24] and hermaphroditism [25]). For many groups, polyploidy has still been presumed to be absent or else resulted in lineages that went extinct, and so are not of contemporary evolutionary significance.

The insects were one such group for which polyploidy was considered unimportant Polyploidy has been documented in 0.01% for described insect diversity [26], but these have been considered spontaneous mutants incapable of passing the polyploid state on. The last major review on insect polyploidy [27] indicated that these individuals occur more commonly for some taxonomic groups and in harsher environments that prompt meiotic failure, but definitively asserted that polyploidy has been inconsequential to insect evolution. However, recent large-scale comparative genomic analyses discovered at least 18 independent WGDs (as well as six other major gene duplication events suggesting polyploidization) across the insect evolutionary tree [26]. This reverses existing theory and suggests that polyploidy is actually an important and pervasive feature of insect evolution.

This new insight incentivizes studying heritable polyploidy in insects to the same level as other groups. However, because of this prolonged lack of interest, there is an absence of equivalent research resources. In particular, a major enigma in polyploid evolution is how specific polyploidization events result in lineage survival and eventual derivation of evolutionary advantage (the "polyploid hop") whereas mostly polyploidization results in extinction [28]. The primary means for investigating this is delineating heritable adaptive mechanisms by comparing non-polyploid and viable polyploid lineages, and by comparing more and less successful polyploid lineages. For the plants and better studied animal groups, this has been

facilitated by the ready induction of viable, reproductive neopolyploids through chemical or environmental shock. There are no known means for doing this with insect species. In fact, the inviability of polypoid *Drosophila melanogaster*, with triploids and tetraploids incapable of establishing stable lineages, has been credited for the misconception of animal polyploid rarity and a consequential lack of study in the first place [3, 18]. At most, whole organismal polyploidy has been induced for a single sterile generation (bumblebees [29]; Lepidoptera [30]).

The Hymenoptera (bees, ants, wasps, and sawflies) are of special interest to polyploid research for several reasons. They reproduce by haplodiploidy, i.e. haploid males develop from unfertilized eggs and diploid females develop from fertilized eggs, so must have mechanisms for maintaining functional "housekeeping" biology for disparate ploidies. They also have numerous species with neopolyploids. However, most are commonly sterile diploid males that mostly arise as a consequence of inbreeding in taxa with a specific form of sex determination (complementary sex determination) [31, 32]. Nevertheless, there are a few species with reproductive polyploids, though typically these are diploid males that then produce sterile triploid daughters (e.g. [33, 34]).

A prominent insect system with readily heritable, viable, and inducible polyploidy is *Nasonia vitripennis* (Chalcidoidea). This species is a parasitoid of blowfly pupae and has been used as a study model for wasp biology for decades [35]. Polyploids appeared spontaneously in Whiting'laboratory lines in the 1950s, and one line has been maintained ever since, known as the Whiting polyploid line (WPL) [36, 37]. In the (WPL), the polyploid state is stably inherited through alternating generations of high fecundity diploid males and low fecundity triploid females. The WPL has long been used to investigate the role of ploidy in sex determination [37–39], and this work has been expanded with polyploids intentionally generated through RNAi knockdown of single genes in the sex determination pathway (*transformer*, *transformer-2*, and *wasp overruler of masculinization*) [40–42]. These knockdowns produce diploid males that are fertile and produce diploid sperm (similar to WPL), leading to neotriploid daughters upon fertilization of haploid eggs. Surprisingly little is known about *N. vitripennis's* polyploid biology and its potential insights for insect polyploidization mechanisms or evolution. We generated a neopolyploid line from a genetically variable strain via *transformer* knockdown, designated it tKDL (for *transformer* knock down line), and compared it to the WPL. We used this rare opportunity of having both a long-established and a newly generated polyploid line to ask some fundamental questions of polyploidization mechanisms and evolution that have always been difficult to study in animals. Regarding the polyploid hop, to what degree can polyploids be reproductively capable? Is polyploid cell size and number regulated by ploidy or sex (or both)? How is global gene dosage regulated in cases of ploidy differences? To do this, we assayed polyploids and non-polyploid counterparts of both lines for 1) male competitive ability for acquiring female mates 2) female fecundity, progeny sex ratio, and progeny polyploid proportion 3) whether cell reduction exists and 4) gene dosage of housekeeping genes across sex and ploidy.

## Materials and methods

### *Nasonia vitripennis* lines and rearing

*Nasonia vitripennis* is a globally distributed parasitoid of blowfly pupae used broadly in genetics, behavioral, and ecological research for decades [35]. It has haplodiploid reproduction; unmated females produce exclusively male offspring from haploid eggs, and mated females produce daughters from fertilized eggs in addition to a proportion of sons. All lines were maintained under conditions with 25˚, 16:8 LD cycle, ~55% RH, with two-week generation cycles on *Calliphora* sp. pupae hosts. The Whiting polyploid line (WPL) was acquired from the J.H.

Werren laboratory (University of Rochester, Rochester, New York, USA) and was maintained in our laboratory for 20 years. This strain is cured of *Wolbachia* endosymbiotic bacteria. It carries two complementary recessive eye-color mutations, *scarlet* (*st*) and *oyster* (*oy*). Virgin triploid females produce red-eyed haploid and diploid purple (wildtype)-eyed males. The diploid purple-eyed males are crossed to females of the red eye mutant line *scarlet*, which are used to recoup triploid females for another breeding cycle (see detailed breeding cycle scheme in [36, 39]). The genetically variable HVRx strain was acquired from B.A. Pannebakker laboratory (University of Wagening, Wagening, The Netherlands). HVRx was founded by mixing several Dutch field lines and genetic variation is maintained by re-hosting four mass culture tubes every generation and mixing hosts several days after oviposition [43]. It has not been cured of *Wolbachia*. This line was used to generate the neopolyploid tKDL line. Untreated mated females from this line were also used to generate control (no descent from injection) individuals for assays. All crosses were done intraline to their WPL or HVRx background.

## Generation of the neopolyploid *tra* KD line (tKDL)

The *tra* KD line (tKDL) was created following the ds *tra* RNA synthesis and female injection protocol of [39]. Maternal knockdown of *tra* results in sex reversal of diploid eggs, i.e. diploid males rather than females. Polyploids (diploidized males) and non-polyploid males (from unfertilized haploid eggs) thus occur in the same generation. To account for descent from the injected females as an independent factor with possible effect on phenotype, assays were conducted considering control non-polyploids from the untreated HVRx population, non-polyploids of the tKDL, and (neo)polyploids of the tKDL. To generate the tKDL line, HVRx virgin females (N = 200) were injected as white stage pupae with 4μg/μl ds *tra* mixed with red dye (to confirm injection) with a FemtoJet microinjector (the F0 generation). They were mated with control haploid males as adults. They were given three hosts, and produced tKDL diploid males from fertilized eggs and tKDL haploid males from unfertilized eggs (the F1 generation). When diploid males, which produce diploid sperm, are mated with diploid females, their female offspring will be triploid. Every generation was outcrossed to the untreated HVRx population to maintain genetic variability (~20 founders or foundresses and five offspring each generation). The fate of the tKDL neopolyploid line was followed to generation F5. S1 Fig indicates individuals used, the crosses to obtain them, and which assays were conducted for each generation. As there were no means to visually distinguish diploid and haploid males in the tKDL they were typed for ploidy using a process combining flow cytometry of heads and fecundity assessment of daughters (diploid daughters will have higher fecundity, reflecting a haploid father, and triploid daughters will have lower fecundity, reflecting a diploid males). The full ploidy-typing process is described in S2 Fig.

## Male mate competitions

Male mate competition experiments were conducted between pairs of tKDL diploid males and haploid males. One experiment competed the diploid with an unrelated haploid male from the untreated HVRx strain. Another experiment competed diploid and haploid brothers from the same tKDL mother. tKDL diploid males were also compared to WPL diploid males. Two types of competition experiments were conducted, single competitions were for one virgin female, and multiple competitions were for 10 virgin females. These females were from the HVRx population, which as a genetically variable background is expected to exhibit strong monandry [44–46] (females mate once and all their female offspring descend from a single father). Male competition experiments were conducted by placing males and females in a single tube for 24 hours. Females were then removed and given three hosts. For the tKDL haploid and tKDL

diploid competitions, the possibility of using two haploids or two diploids could not be guaranteed a priori. Therefore, post-competition, males were individually mated to single HVRx females, to generate daughters for the daughter-typing component of the ploidy-typing process, and subsequently stored at -20˚C for the flow cytometry step (S2 Fig). For all competitions, the male mate of female was determined by typing the ploidy of daughters (diploid daughters indicated a haploid father, triploid females a diploid father) (described in S2 Fig). For the tKDL diploid and WPL diploid competitions, daughters were instead hosted as virgins. Those that had male offspring with red eyes indicated that their mother mated with the WPL male, and those that did not have male offspring with red eyes had mothers that mated with the tKDL male. Females from competitions that produced all male offspring were presumed to have not mated and were excluded from analyses, as were the tKDL competitions that were found to have used two males of the same ploidy.

## Female fecundity and offspring ploidy distribution

To bypass any knockdown effects and to increase sample size, we examined tKDL F4 generation instead of the F2 generation (S2 Data). We scored progeny size for virgin and mated diploid control females, F4 tKDL diploid and triploid females, and WPL triploid females to assess the influence of polyploid background on fecundity. For the mated females, we also assessed progeny sex ratio. In the case of the triploid females, progeny ploidy distribution was also scored, as eggs of triploid females can be haploid or diploid. Females of the mated assays were given 24 hours to mate with a control haploid HVRx male. All females were then given three hosts. All progeny were counted for each female 14 days later, and in the case of the mated females, sexed. For the virgin and mated triploid F4 tKDL triploid females, entire families were scored for ploidy using flow cytometry (excluding larvae, which cannot be typed for ploidy with flow cytometry) (S1 Fig).

## Wing cells and cell reduction mechanisms

To investigate the possible existence of cell reduction mechanisms in *Nasonia* polyploids (and as a case study for cell reduction in invertebrate and insect polyploids), we examined wing cells for both sexes across all ploidy levels for both the WPL and the tKDL. We counted setae (wing hairs), as each hair corresponds to single cell (following [47]) (S3 Data). The adult right forewing was removed for each individual and mounted on a glass slide using clear nail polish. Using the Motic Images Plus 2.0ML program, wings were imaged with a Moticam 2000 camera attached to a Car Zeiss SV6 microscope. In Photoshop CS6 (64 bit) a 0.25 mm$^2$ square subsection below the cross-vein was consistently sampled for each specimen (Fig 2), and setae counted with the count tool. For the WPL, females of the *scarlet* line used for maintaining the line represented control diploid females. For the tKDL diploid male measurement, ≤20% of individuals might have actually been haploid (based on typical male production of mated *N. vitripennis* females [48]) because haploids could not actually be sorted from this class. However, significant differences among groups could be attributed to the majority presence of diploids. For each group, N = 5 randomly selected individuals were also scored for setae count across the whole right forewing using the count tool in Photoshop.

## Gene dosage of housekeeping genes *Ak3* and *ef1α*

To assess the effect of ploidy and sex on gene expression, we quantified the absolute expression of two housekeeping genes *Adenylate kinase 3* (*Ak3*) and *elongation factor 1 alpha* (*ef1α*) for all backgrounds (N = 5 each). We followed the protocol of Dalla Benetta et al., [49] with the following modifications. In brief, RNA was extracted from heads and abdomens separately. Each

wasp's head was individually placed into a 1.5 ml Eppendorf tube. The same was done for abdomens. Thoraxes were discarded due to likely endopolyploidy [50]. Samples were immediately frozen at -80˚ C, extracted using TriZol (Invitrogen, Carlsbad, CA, USA), and cDNA conversion performed according to manufacturer's instructions. Five technical replicates for each sample type were performed to control for pipetting error. qPCRs were run for *Ak3* and *ef1α* (primers in S1 Table) on the Applied Biosystems 7300 system. Reactions were first performed at a 5x dilution (Fig 2, S1 Data) in the event of large amount of RNAi producing asymptotic readings instead of accurately measuring the absolute expression levels. Analyses were then also run for 50x dilutions to test for consistency (S2 Data). The qPCR reactions were run with 3 min of activation phase at 95˚C, followed by 35 cycles of: 15s at 95˚C, 30 s at 56˚C, and 30 s at 72˚C. Note that in typical qPCR reactions these two genes are used as reference for RNA quantity, but in this assay we use the amplification amount as a direct measure for gene expression level as function of ploidy and sex.

## Statistical analyses

All data are reported as means with standard deviation. Statistics were done in SPSS Statistics 25 [51] and R [52]. No datasets met normality or homogeneity of variance (Shapiro-Wilks Brown-Forsythe tests), so Kruskal-Wallis rank sum tests were used to compare groups. Post-hoc Dunn's multiple comparison tests with a Bonferroni correction were performed to test for differences between paired groups. Binomial tests were used to assess mate competitions with null hypotheses that competing male types had equal mate competition ability in every competition. An additional general linear mixed model was used to determine how likely a female was to mate with each male type in each competition type (S1 Table). The qPCR results were analyzed using the LinReg PCR software, with absolute expression normalized to the program's calculation for $N_0$ concentration [53, 54]. A general linear model was also used to analyze these results with extraction set, and individual set as random effects; and ploidy, sex, and background (control, tKDL, or WPL) set as fixed effects. A Satterthwaite approximation and estimation of robust variance was used to account for low sample size and non-normal distribution.

## Results

### Male mate competition

Like most parasitoid wasps, *Nasonia* have female mate choice and females typically only mate once [44, 45]. In other hymenopteran studies [55], including that on WPL [36], diploid male mate competition ability is equal to haploids. To compare our lines to this existing trend in polyploid mate competition research, we conducted competitions between diploid males of the WPL and tKDL against haploid males of an untreated control line, and haploid-diploid brother pairs of the WPL and tKDL. We also competed diploid males of WPL against diploid males of tKDL. We did this for competitions for single females and multiple (N = 10) females. In the single female mate competition trials the successful male was scored as the one that sired offspring, and in the multiple female trials, the successful was the male that sired offspring with more females overall. Competitions were performed a priori without knowing the ploidy of the tkDL males, which was checked after using flow cytometry (S1 Fig). The tKDL diploid males were consistently outcompeted by haploids of both the control background of the untreated HVRx population, and their own haploid brothers from the same tKDL background (Table 1). In competitions for the single female haploid males, the tKDL diploid male only won 9 out of 33 trials (P = 0.014) when competing with a control haploid male, and 7 out of 21 competitions against a haploid tKDL brother (P = 0.095, binomial test). For multiple females, the tKDL diploid male only won 1 out of 22 competitions against a control haploid

**Table 1. Male mate competitions.** Males competition pairs were given either a single or multiple (N = 10) virgin female(s), and winners were determined through off-spring scoring of daughter fecundity or flow cytometry. The winning male type (more trials won) is marked with an asterisk (*). P-values are for binomial tests.

| Competition | Male type 1 | Trials won | Male type 2 | Trials won | *P*-value |
|---|---|---|---|---|---|
| single female | control haploid* | 24 | F1 tKDL diploid | 9 | 0.014 |
| multiple females | control haploid* | 22 | F1 tKDL diploid | 1 | <0.001 |
| single female | F1 tKDL haploid* | 14 | F1 tKDL diploid | 7 | 0.095 |
| multiple females | F1 tKDL haploid* | 22 | F1 tKDL diploid | 6 | 0.095 |
| single female | control haploid* | 3 | F5 tKDL diploid | 1 | 0.313 |
| multiple females | control haploid* | 8 | F5 tKDL diploid | 2 | 0.001 |
| single female | F5 tKDL haploid* | 10 | F5 tKDL diploid | 4 | 0.18 |
| multiple females | F5 tKDL haploid* | 9 | F5 tKDL diploid | 1 | 0.022 |
| single female | WPL diploid | 4 | F1 tKDL diploid* | 11 | 0.118 |
| multiple females | WPL diploid | 4 | F1 tKDL diploid* | 9 | 0.267 |

male (P<0.001, binomial test), and 6 out of 28 competitions against a haploid tKDL brother (P = 0.095, binomial test).

As these experiments were done in the F1 generation immediately following RNAi knock-down, we considered that RNAi carry over effects could be influencing results. So eliminate this possibility and to test for heritable effect, we repeated these experiments in the F5 generation. Flow cytometry indicated that many trials had to be discarded because a haploid tKDL male rather than a necessary diploid tKDL male was used. However, even for the resultant F5 smaller sample size, a pattern of tKDL diploid male inferiority against haploids held. For competitions for a single female, the F5 tKDL male only won one out of four single competitions against a control haploid male (P = 0.313, binomial test), and four out of 14 single competitions against a F5 tKDL haploid brother (P = 0.180, binomial test). In competitions for multiple females, the F5 diploid male only won two out of ten competitions against a control haploid male (P = 0.001), and one out of ten competitions with a F5 haploid tKDL brother (P = 0.022). The low preference of females for the tKDL diploid male is reflected in a general linear mixed model analysis (GLMM), with females being 2.3–3.3 times less likely to mate with diploid males versus haploid males in the F1 generation, and 2.4–22.6 times less likely to mate with diploid males than haploid males in the F5 generation (Table 2).

**Table 2. General linearized mixed models (GLMM) to test whether females were more likely to mate with one type of male in mate competition experiments.** Trial number was set as a random effect. A multinomial logistic regression with a generalized logit link and a Satterthwaite approximation and estimation of robust variance were used to account for low sample size and non-normal distributions. The exponential coefficient gives the number of times females are more likely to mate with the superior mate competitor type (the haploid in every competition, or tKDL in the diploid-diploid competitions). Significant results are indicated with an asterisk (*).

| Competition | BIC | Co-efficient | Exp. Coefficient | *P*-value |
|---|---|---|---|---|
| Intercept | | | | |
| control haploid vs. F1 tKDL diploid (single)* | 149,662 | 0.982 | 2.669 | 0.019 |
| control haploid vs. F1 tKDL diploid (multiple)* | 10,358 | 1.188 | 3.282 | <0.001 |
| F1 tKDL haploid vs. F1 tKDL diploid (single) | 93,689 | 0.818 | 2.265 | 0.271 |
| F1 tKDL haploid vs. F1 tKDL diploid (multiple)* | 1,160,746 | 0.881 | 2.414 | <0.001 |
| control haploid vs. F5 tKDL diploid (single) | 16,804 | 1.186 | 3.723 | 0.463 |
| control haploid vs. F5 tKDL diploid (multiple)*[1] | 470,536 | 3.119 | 22.625 | <0.001 |
| F5 tKDL haploid vs. F5 tKDL diploid (single) | 63,424 | 0.854 | 2.384 | 0.284 |
| F5 tKDL haploid vs. F5 tKDL diploid (multiple)* | 459,686 | 2.997 | 20.030 | 0.002 |
| F1 tKDL diploid vs WPL diploid (single) | 65,507 | 1.889 | 6.613 | 0.076 |
| F1 tKDL diploid vs WPL diploid (multiple)* | 276,730 | 1.502 | 4.490 | 0.038 |

[1]Trial number was removed as a random effect because a lack of variation prevented a positive definite Hessian matrix

To specifically test for differences between polyploidization background (neopolyploid and generated tKDL versus long-existing and spontaneous WPL), we competed F1 tKDL diploid males against WPL diploid males (Table 1). Despite the tKDL diploid males performing poorly against haploids, whereas WPL diploid males had equal mate competition ability to haploids in a previous study [36], the F1 tKDL diploid males outperformed the WPL diploid males. In competitions for a single female, the F1 tKDL diploid male won 11 out of fifteen competitions against the WPL diploid male (P = 0.118, binomial test), and in competitions for multiple females, the F1 tKDL male won nine out of 13 multiple competitions (P = 0.267, binomial test). Although sample sizes were small and so not significant, tKDL males were 4.5–6.6 times more likely to mate with females than WPL diploids (Table 1). In addition, fewer females mated in the tKDL diploid-WPL diploid multiple mate competitions (on average, 4.3 females) compared to the tKDL haploid-tKDL diploid competitions (7.70–9.25 females) (P<0.05, Kruskal-Wallis test and Dunn's test). In short, tKDL diploid males appear to be inferior competitors against haploids, but are superior competitors to WPL diploid males, and in general diploid males are less able to induce mating acceptance from females than haploids.

## Female fecundity

The high fecundity of *N. vitripennis* polyploid males is known from the WPL [37], with WPL diploid males having the same number of offspring as haploids with a single mate [36]. We assessed the progeny size of virgin and mated WPL triploid females, and tKDL diploid and triploid females, under standard culturing conditions. We further measured the progeny sex ratio of mated females, as females vary in the number of eggs they fertilize based on various environmental and genetic factors [45–49]. We also assessed the proportion of progeny that were polyploid for virgin and mated females for both WPL and tKDL. We did this to investigate the degree by which polyploidy impacts *N. vitripennis* fecundity, which is a major challenge to the establishment of polyploid lineages. These data are summarized in Table 3.

For WPL triploids, the mean progeny size was low, 3.64 ± 3.32 for virgins (N = 49) and 2.34 ± 2.43 for mated females (N = 50). This conforms to expectations of high aneuploidy. In contrast, diploid *scarlet* females used to maintain the line produce 60–90 offspring. The mean progeny size of tKDL triploids was higher than WPL triploids for both tKDL triploid virgins (12.40 ± 4.94, N = 50) and mated females (20.77 ± 8.49, N = 47). As expected, tKDL triploid females had lower virgin progeny sizes than control diploids (97.73 ± 19.59, N = 45) and tKDL diploids (69.18± 18.00, N = 49). The tKDL triploid females also had lower mated progeny sizes than the control diploids (59.34 ± 20.70, N = 47) and tKDL diploids (74.78 ± 18.24, N = 49). All WPL triploid versus tKDL triploid differences progeny size differences were significant, as were all tKDL triploid differences with the control and tKDL diploids (P<0.001 Kruskal-Wallis test, P<0.05 Dunn's test). These results reflect a tKDL triploid progeny size 3.4 times (virgin) and 8.9 times (mated) larger than that of WPL triploids.

A typical *Nasonia* progeny sex ratio (male offspring/total offspring) of about 25% male [56] was consistent among the mated tKDL triploids (0.214 ± 0.174, N = 47), control diploids (0.233 ± 0.178, N = 47), and the tKDL diploids (0.264 ± 0.217, N = 49) (P = 0.448, Kruskal-Wallis test). In contrast, mated WPL triploids (N = 50) skew heavily towards male offspring, with a mean progeny sex ratio of 0.89 ± 0.26, which is significantly higher than the tKDL triploid female (P<0.001, Dunn's test). This indicates that the tKDL triploids not only produce more viable euploid eggs than the WPL, but also that these eggs are more amenable to fertilization and female production.

Whole broods of virgin (N = 31) and mated tKDL (N = 47) triploid females were typed for ploidy using flow cytometry to determine if this line has higher polyploid production than

**Table 3. Female fertility.** Progeny size is reported for all females. Progeny sex ratio is reported for mated females. Progeny polyploid proportion is reported for triploid tKDL females. All data are means ± standard deviation. NA indicates that data was not collected. Relevant significant pairwise differences are discussed in the text.

| Female type | Total offspring | Offspring sex ratio (m:total) | Polyploid offspring (%) |
|---|---|---|---|
| virgin control diploid (N = 45) | 97.73 ± 19.59 | NA | NA |
| mated control diploid (N = 47) | 59.34 ± 20.70 | 0.233 ± 0.178 | NA |
| virgin WPL triploid (N = 49) | 3.64 ± 3.32 | NA | NA |
| mated WPL triploid (N = 50) | 2.34 ± 2.43 | 0.89 ± 0.26 | NA |
| virgin tKDL diploid (N = 49) | 69.18 ± 18.00 | NA | NA |
| mated tKDL diploid (N = 48) | 74.78 ± 18.24 | 0.264 ± 0.174 | NA |
| virgin tKDL triploid (N = 50) | 12.40 ± 4.94 | NA | 31.9 ± 22.4 |
| mated tKDL triploid (N = 47) | 20.77 ± 8.49 | 0.214 ± 0.174 | 36.6 ± 1.91 |

WPL. The mean polyploid percentage (polyploid offspring /total offspring) was 31.9 ± 22.4%, and 36.6 ± 1.91% for virgin and mated females respectively. This is not different from the 25% polyploid proportion reported for WPL [37, 57], but lower than the 50% proportion expected from random segregation.

## Cell size and reduction

Setae count for subsections and whole wings of WPL and tKDL indicated a difference in cell number and size for polyploid backgrounds (Table 4). For WPL, for both the sampled subsections and whole wings, there was a clear progression of fewer, larger cells for higher level ploidy, and females having fewer and larger cells than males (Fig 1B). Haploid WPL males (subsection: 189.19 ± 38.22, N = 47; whole wing: 506.4 ± 67.12, N = 5) averaged significantly more cells than diploid WPL males (subsection: 157.20 ± 38.22, N = 25; whole wing: 394.4 ± 45.55, N = 5). WPL Females had fewer and larger cells than males for subsections, but more for whole wings, as female wings are larger. Between WPL females, diploid *scarlet* (WPL) females had more cells (subsection: 46.71 ± 5.98; N = 34; whole wing; 902.2 ± 65.44, N = 5) than triploid WPL females (subsection: 35.63 ± 5.28; whole wing: 709.6 ± 59.6, N = 5).

In contrast, there were no clear progression of cell number or size differences for ploidy levels or sex for tKDL (Fig 1C). For the subsection sampling, haploid control males (69.73 ± 16.83, N = 87) had significantly fewer cells than the tKDL diploid (95.93 ± 15.39 N = 79), the opposite of what would be expected in the case of polyploid cell reduction.

**Table 4. Subsection and whole wing setae count.** Pairwise comparisons are Mann-Whitney U tests, multiple comparisons are Kruskel-Wallis tests. An asterisk (*) indicates the group that has significantly more cells.

| Wasp type | subsection | P-value | whole wing (N = 5) | P-value |
|---|---|---|---|---|
| haploid WPL male (N = 47) | 189.19 ± 38.22* | 0.005 | 506.4 ± 67.12* | 0.047 |
| diploid WPL male (N = 25) | 157.20 ± 42.33 | | 394.4 ± 45.55 | |
| diploid WPL female (scarlet) (N = 34) | 46.71 ± 5.98* | <0.0001 | 902.2 ± 65.44* | 0.009 |
| triploid WPL female (N = 32) | 35.63 ± 5.28 | | 709.6 ± 59.65 | |
| haploid control male (N = 86) | 69.73 ± 16.83 | <0.0001 | 580.4 ± 29.29 | 0.095 |
| diploid tKDL male (N = 79) | 95.93 ± 15.39* | | 485.8 ± 77.10 | |
| diploid control female (N = 20) | 49.25 ± 5.41 | 0.053 | 1417.4 ± 85.69 | 0.069 |
| diploid tKDL female (N = 21) | 54.38 ± 5.72 | | 1361.2 ± 97.22 | |
| triploid tKDL female (N = 55) | 50.8 ± 6.33 | | 1257.8 ± 72.46 | |

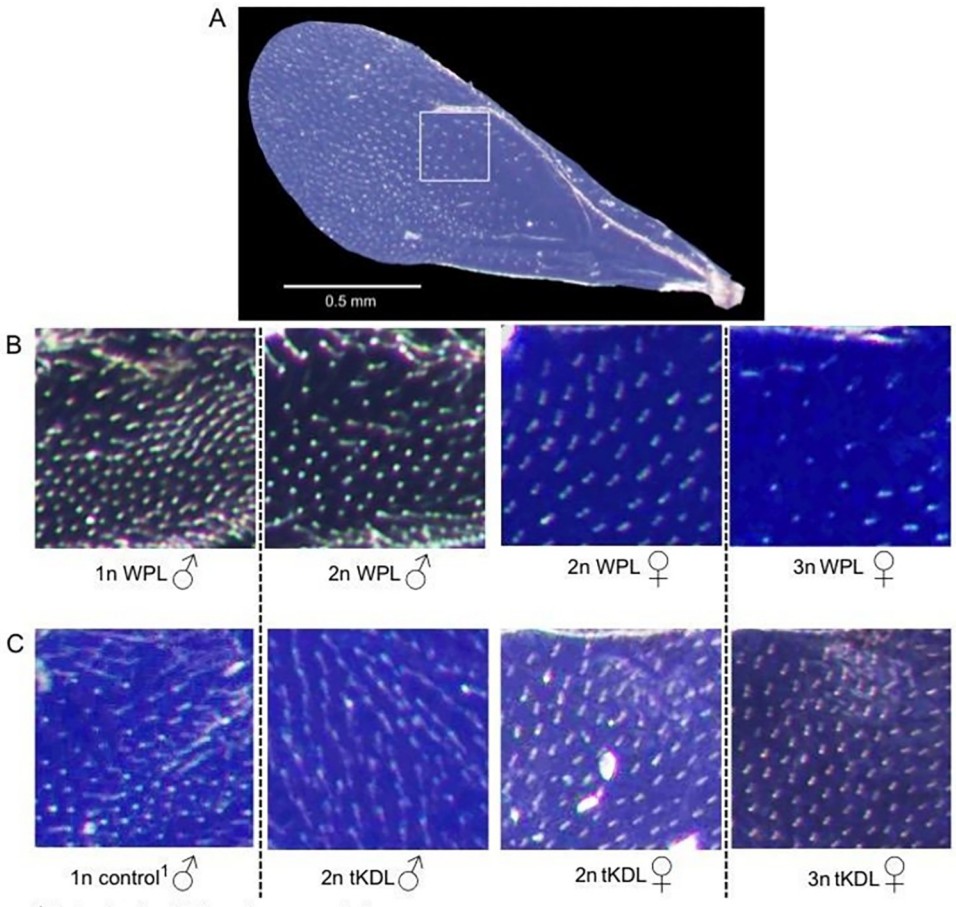

**Fig 1. Cell reduction exists in the long-established Whiting polyploid line (WPL) but not the neopolyploid** *transformer* **knockdown line (tKDL).** (A) 0.25 mm$^2$ subsamples of the right forewing were taken using the single cross vein as a landmark to ensure consistency (specimen pictured is a 3N WPL female). As each seta (wing hair) corresponds to a single cell, greater distance between setae indicates larger cells. (B) In the Whiting polyploid line, cells become progressively larger with ploidy level, and as wings are similarly sized within sex for polyploids and non-polyploids, this suggests cell number reduction mechanisms at work (C) In contrast, in the tKDL knockdown line, cell size remains the same across sex and ploidy levels, suggesting that cell number reduction mechanisms only evolve over time, or that *tra* has a role in cell and body size regulation that has been lost in these specimens. Dotted lines for (B) and (C) indicate separation of ploidy levels. For quantitative values, see Table 4.

However, there was no significant difference between haploid control (580.4 ± 29.29, N = 5) males and tKDL diploid males (485.8 ± 77.10) for whole wing setae count. Again, females had fewer subsection cells males and more cells overall for larger whole wings. However, there were no significant differences among the control diploid, tKDL diploid, and triploid tKDL females for subsections (control diploid females: 54.38 ± 5.72, N = 20; tKDL diploid females: 49.25 ± 5.41, N = 21; tKDL triploid females: 50.82 ± 6.33, N = 55) or whole wings (control diploid females: 1417.4 ± 85.69, N = 5; tKDL diploid females: 1361.2 ± 97.22, N = 5; tKDL triploid females: 1257.8 ± 72.46, N = 5). (P-values for all comparisons are reported in Table 4). It should be noted there is large overlap in values among the groups (as is reflected by the large standard deviations), indicating that even in WPL there is no clear segregation in setae count between polyploids and non-polyploids. These data suggest cell size increase and cell number reduction in one polyploid background (WPL) but not for another (tKDL).

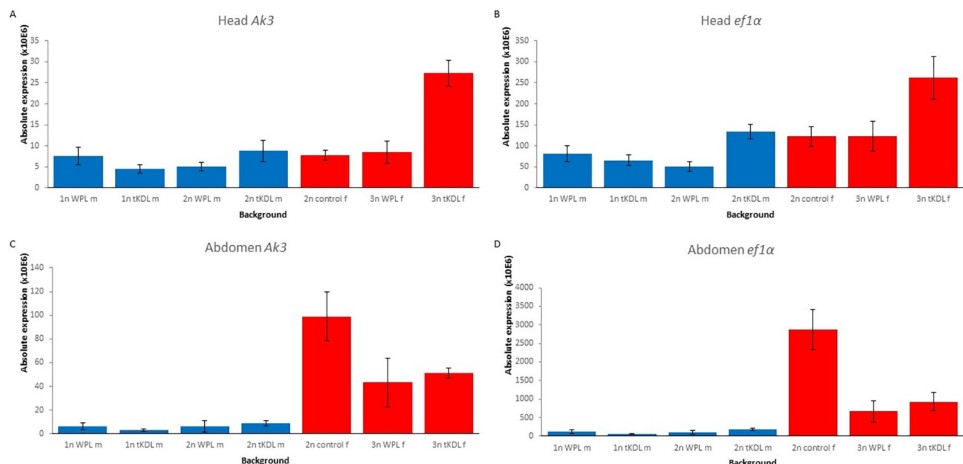

**Fig 2. qPCR results for absolute gene expression of housekeeping genes *Ak3* and *ef1α*.** Absolute expression and standard error for three replicates for 5x dilution of cDNA for (A) *Ak3* in heads (B) *ef1α* in heads (C) *Ak3* in abdomens and (D) *ef1α* in abdomens. Blue indicates male background, and red indicates female background. Control indicates non-injected individuals of the strain used to generate the *transformer* knockdown line (tDKL). If the prevalent hypotheses (expression scales to ploidy and sex-linked dosage mechanisms do not exist in Hymenoptera) were true expression would simply be 1x, 2x, and 3x for haploids, diploids, and triploids, respectively. Instead, expression is consistent for 1n and 2n males (with the exception for the head expression of the 2n *tra* KD males, possibly because these individuals would have normally developed into females without deactivation of the feminization process through *tra* knockdown). These data suggest that sex-linked expression conservation may be a mechanism to modulate dosage changes from polyploidization. (These patterns are also consistent for a 50x dilution).

## Gene expression levels

Generally, the absolute expression of the housekeeping genes *Ak3* and *ef1α* for a 5x dilution of cDNA (Fig 2) was lower in males than females. This applied to both the head and abdomen, and for both the WPL and the tKDL backgrounds (Fig 2). Results for a 50x cDNA dilution (S2 Data) replicated all these patterns, demonstrating that the 5x dilution results were not biased by upper limits of expression detection. There was no pattern of an increase in expression correlating to increased ploidy level. For *Ak3* and *ef1α* head expression, males of almost all backgrounds (WPL haploid, WPL diploid, control haploid and tKDL haploid males) resembled each other strongly. However, tKDL diploid male head expression was female-like (like the diploid and triploid females). As these males arose from diverted female development through *tra* knockdown, it is possible they are deficient in male character relative to typical haploid males. Overall, there was more variation in female head expression, which tended towards being higher than the males. However, for *Ak3*, in diploid control, tKDL diploid, diploid (*scarlet*) WPL, and triploid WPL females the expression was as low as the haploid WPL male's (Fig 2A). A clearer separation of lower male values versus higher female values was observed for the head expression of *ef1α* (Fig 2B). Abdominal *Ak3* and *ef1α* expression was consistently low for WPL haploid, WPL diploid, tKDL haploid, and tKDL diploid males (Fig 2C), and high but variable for control diploid, WPL diploid (*scarlet*), WPL triploid, tKDL diploid, and tKDL triploid females (Fig 2D). These data show that in two polyploid backgrounds, gene expression does not scale to ploidy, but rather is dependent on sex.

## Discussion

Although polyploidy was once thought of as an evolutionary dead end [6, 7], it is now recognized as a major force for success by underlying gene network diversification, increased hardiness, expanded geographic range, and accelerated speciation [5, 8, 16, 21]. Despite these

advantages, immediate detriments of polyploidization are highly challenging, and little is known about how initial barriers to polyploid establishment are compensated [28, 29, 53, 58]. It is often suggested that variation in polyploid mechanisms must impact the severity of initial phenotypes and their fitness, and thus determine likelihood of survival versus extinction. However, few means to experimentally induce or maintain polyploidy in animals has limited its study [9, 58, 59].

We studied a long-maintained polyploid line (WPL) that arose spontaneously in the 1960s through an unknown pathway and a neopolyploid line generated through knockdown of a single sex-determining gene (*Nvtra*) target in a genetically variable background (tKDL) in the parasitoid wasp *N. vitripennis*. We present empirical evidence that there can be major variation in polyploid phenotypes within a single animal species, demonstrating that specific polyploidization pathway can correspond to different degrees of polyploid detriment. Although some outcomes may be specific to the hymenopteran insect group, it reflects how polyploid-induced phenotypes can correspond to the survival likelihood of a lineage.

## Male-female mating interactions differ between and among polyploid backgrounds

In our study, neopolyploid males were poor competitors against haploids for female mates (Tables 1 and 2). This was observed for both the F1 and the F5 generation, indicating a heritable phenotype. Our results diverge from the near-consensus of all other hymenopteran studies [55] (including WPL [36] that diploid males have equal mating success to haploid counterparts. The tKDL diploid males' lesser ability to compete for females adds complexity to the current hymenopteran "diploid male vortex" narrative. For Hymenoptera in general it has been assumed diploid males are not inferior at mating success, so accelerating extinction for species with sterile diploid males, particularly those that arise from inbreeding as in species with complementary sex determination (CSD) [31–33, 60–62]. For CSD, hemizygotes for a *csd* locus or loci are haploid males and heterozygotes are diploid females, but homozygotes are typically reproductively impaired males. It is unclear what proportions of Hymenoptera are CSD versus non-CSD species. Here, we provide an example (for any hymenopteran) that diploid males of different backgrounds of a single species can be highly contrasting in their success with females. Additionally, despite performing poorly against haploids of their own background, the tKDL diploid males were superior mate competitors against the WPL diploids. Therefore, no general effects of polyploidy can be inferred, without considering various initial polyploidy events. Consequently, polyploidization cannot be considered as a single event but is dependent on the duplicated genome characteristics. Apparently, not all genomes are equally suited to survive diploidization.

The cause for disparities in tKDL and WPL diploid male mate competition ability are currently unknown. Although more study is needed, tKDL diploid males are not deficient in courtship or copulatory behaviors, and females do not reject them when they have no other mate choice. The fecundity of *N. vitripennis* diploid males is also similar to haploid males for both WPL [36] and tKDL [60, 63], so female preference for haploids does not seem to be based on reproductive potential. The tKDL diploid males also have no obvious physical impairment (as in the only other known case of hymenopteran diploid male mating failure, in which the male was too large for the female [64]). Interestingly, in the multiple competitions, male rejection was higher when only diploids were present, with female mating rate being lower in the tKDL diploid and WPL diploid competitions compared to the tKDL diploid versus haploid competitions. This suggests that even if they themselves have impaired attractiveness, diploid males may benefit from ambient haploid cues increasing female receptivity, whether

behavioral, mechanical or, chemosensory. This could also partially account for the WPL diploids' success against their own haploids [36].

**Aneuploidy effects vary between polyploid backgrounds.**   As is typical for most polyploids, triploid female hymenopterans are sterile due to a high frequency of aneuploid gametes [5, 33, 34]. The triploid females of the WPL have highly reduced fecundity due to aneuploidy, apparent from the presence of many shriveled eggs in oviposited hosts [37]. It was therefore highly surprising that the neopolyploid tKDL triploids exhibited 3-10x higher fecundity than the long-maintained WPL triploid females (Table 3). This is the first known case of extreme variation in triploid female fecundity within a single species. Thus, tKDL triploid females seem to have an ability to circumvent aneuploidy that the WPL triploids lack.

The mechanism for aneuploidy circumvention in tKDL triploids is unknown, but one possibility is biased meiotic segregation. Uneven ploidy levels (e.g. triploid) are harder to establish than even ploidies (e.g. tetraploid) due to aneuploid gametes [5]. However, this may be alleviated by a parent-of-origin effect on chromosome segregation, i.e. during triploid meiosis the two sets from the diploid parent segregate together and rarely pair with the set from other parent (although this effect may fade with recombination). There is some evidence for this in plants [65–67]. A future study with molecular or morphological markers can test for whether there is biased segregation for polyploidized chromosome sets, giving insight on a potential adaptive mechanism against neopolyploid infertility.

Notably, mated WPL triploid females had fewer offspring than virgins and produced few daughters. The mated tKDL triploid females produced larger and female-biased broods, in line with typical *Nasonia* biology [56]. This suggests that in tKDL (but not the WPL) polyploids, mating induces increased offspring production and egg fertilization.

## Cell reduction varies between polyploid backgrounds

The WPL but not the tKDL appears to have a sex-linked cell reduction mechanism. There are fewer, larger cells in higher-level ploidy individuals (Fig 1, Table 4). Consistent with most hymenopterans (A. Thiel, pers. comm), polyploids and non-polyploids do not differ much from each other in overall body size for tKDL or WPL [63]. Therefore, it does not seem that a cell reduction mechanism is required to avoid polyploid gigantism, making its function for WPL unclear. It could simply be that cell reduction is a later stage polyploid phenotype, and so is present in long-maintained WPL but not neopolyploid tKDL. Whereas polyploid body size and its relationship to cell number and size has been sparsely studied outside of vertebrates [4, 68, 69], this is the first example of cell reduction varying drastically within an animal polyploid.

It should be noted that these inferences are clouded by *tra* having a known role in body size regulation (in *Drosophila* [70, 71]). The absence of cell reduction in tKDL could be an artefactual result of deactivated *tra*, and not a difference in polyploid phenotype between polyploid backgrounds per se. General corruption of *N. vitripennis tra*, whose best-known role is sex determination through activation of the feminization pathway, is apparent from the observation of occasional gynandromorph offspring from the dsRNA injected female. As other genes that can be targeted to create *N. vitripennis* neopolyploids [41, 42] do not have a known role in body size, additional study of knock down lines and longer-term study of neopolyploids will elucidate how common cell reduction mechanisms are in polyploids, and whether they evolve over time. Ideally, more tissues should be sampled to investigate cell reduction in *Nasonia* polyploids. Unfortunately, this is complicated by some tissues having endopolyploid (multinucleated) cells; for example, nearly all hymenopterans have male endopolyploid thoracic tissues [50]. These cells may be larger, preventing accurate comparison between non-polyploid and

polyploid individuals. Cell-counting accuracy of 3D organs is also difficult due to incomplete marker penetrance or disassociation of individual cells. A good candidate for developing protocols for further cell reduction investigation are the brain cells, which consistently and accurately reflect the individual's ploidy [72]. A reference atlas for the *Nasonia* brain has recently been published [73], so it should be possible to *i.e.* use exact neuron count to more fully assess cell reduction in *Nasonia* polyploids.

## Gene dosage is controlled by sex and does not scale to ploidy

How a neopolyploid organism copes with expression changes following a sudden increase in genomic material is one of the biggest questions of polyploid biology [5, 39, 74]. An intuitive assumption would be that gene transcript number scales directly with ploidy to maintain balanced gene networks (e.g. 1x haploid, 2x diploid, 3x triploid) [5]. However, there has been little evidence to support this [75, 76]. Additionally, sexual dosage compensation mechanisms vary between organismal groups (e.g. mammals versus birds versus insects[77]) and it is not clear whether dosage compensation mechanisms exist in haplodiploids to accommodate disparate ploidy levels inherent to the sexes, despite their lack of sex chromosomes [2, 50, 78]. We examined WPL and tKDL to ask 1) is absolute gene dosage directly related to ploidy 2) do hymenopterans have sexual dosage compensation 3) if they do, are they disrupted by polyploidization and 4) do dosage effects vary across polyploid backgrounds?

In our study we observed in both WPL and tKDL a strong pattern of sex-linked dosage conserved against polyploidy (Fig 2, S2 Data), which has not been previously observed in any other hymenopteran species. That we studied absolute expression of two housekeeping genes tentatively implies a general mechanism across genes without sex-specific function. Interestingly, throughout Hymenoptera (except for basal-most Xylidae), endopolyploidy (diploidy) is high in body tissues of haploid males, purportedly to match the metabolic and mechanical abilities of diploid females [50]. This would imply that the haploid genome cannot just double its expression. Yet, in our study, male diploidy does not increase expression (in the head or abdomen), calling into question if they differ fundamentally from (diploid) body cells, and if so, how and why.

Although there have been a few studies on diploid male ants [79, 80], this study focuses on polyploid hymenopteran expression to consider triploid females. We uncovered consistent sex-conserved dosage for both the long established and the neopolyploid backgrounds despite their many contrasting polyploid phenotypes, suggesting an ingrained mechanism that may act as a buffer against the genomic shock of ploidy changes. It may be a factor in how there are additional forms of aberrant ploidy in *Nasonia* such as haploid [79] and tetraploid females [37] that are viable and not seriously impaired outside of reproductive function. Although more in-depth studies are needed, particularly across whole transcriptomes and with rigorous sampling standardization [75], our results point at the existence of sex-linked mechanisms that buffer against would-be dosage altering effects of polyploidization.

## Phenotypic variation bridges the polyploid hop and *Nasonia vitripennis* as an experimental animal polyploid model

We synthesize our results to introduce the idea that different polyploidization events can reflect extreme phenotypic variation in polyploid phenotypes, with some allowing the polyploid hop to be bridged (Fig 3). In an evolutionary context, there is a gradient of outcomes. Highly deleterious pathways resulting in an "evolutionary dead end" are the most common. However, it has long been hypothesized without experiment-driven proof that there are pathways that have fewer impacts on fecundity and cause fewer problems of epigenetic disruption

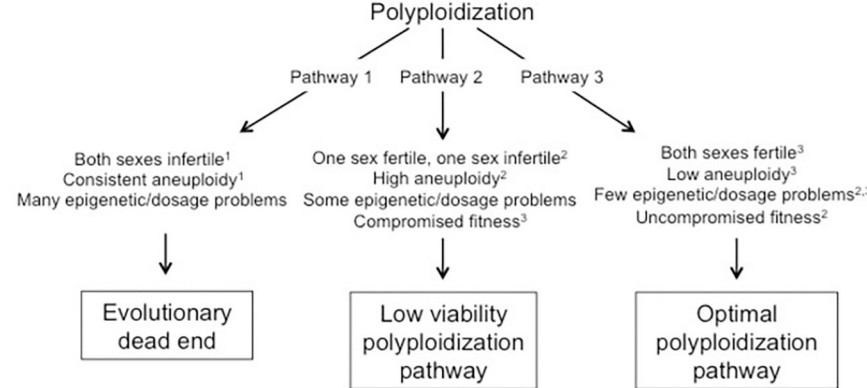

**Fig 3. Outcomes of hypothetical alternate polyploidization pathways.** An organism can become neopolyploid through multiple pathways representing a gradient of least likely to result in polyploid lineage establishment (an evolutionary dead end) to most to likely (the hypothetical optimal polyploidization pathway). However, one polyploid pathway can simultaneously result in some neopolyploid phenotypes that are more conducive to survival and success, and others that are not. Both WPL and tKDL serve as examples, but for different traits.

to allow establishment of polyploid lineages [5, 68]. Although our study concerns a system with haplodiploidy (~20% of animal species [80]), it lends possible insight in the prevalence of WGDs in the ancestry of many eukaryotes.

Polyploidization occurs more often in invertebrates than vertebrates [5, 68, 81], but animal polyploidy research has been heavily biased towards fish and amphibians [18, 23, 69]. This is possibly due to the popularity of some of these vertebrate species as models, but this ironically translates to a striking dearth of knowledge of polyploid evolution in invertebrates, the taxa for which it is most common. Furthermore, just as there has been taxonomic bias in favor of plants versus animals in polyploid research due to easier means of study and a misconception about relative evolutionary and applied importance [18, 69], there has been a parallel situation in hymenopteran polyploid research. Most of this work has been done in CSD species, possibly because polyploidy is readily inducible (i.e. through inbreeding) and is readily detected. However, non-CSD parasitoid wasps comprise most of the commercially important hymenopterans [82] and the majority of hymenopteran diversity (including the *Nasonia's* superfamily Chalcidoidea, which by itself may account for 6% of all animal diversity on Earth [74, 83].

The types of polyploid questions investigated in other animal systems are also perhaps illustrative of the specific advantages and limitations of neopolyploid research in *Nasonia*. For example, both the fish genus *Carrasius* and the frog lab model *Xenopus* also exhibit a range of ploidies. *Carrasius* specifically has several modes of sexual and asexual reproduction and varying degrees of fecundity, and work has been done to link these factors to ploidy [84]. *Xenopus* has the widest range of ploidy levels in tetrapods (diploid to dodecaploid), originating from species hybridization events (allopolyploidy). It has been used for example to study the evolution of subgenomes over evolutionary time [85]. Unlike *Crassius* and Xenopus, *Nasonia* polyploidy has not been recorded in nature, so it cannot be used to study real-world evolution of polyploid population dynamics (although artificial lab populations are potentially possible). Furthermore, there is no evidence that *Nasonia* allopolyploids are possible, although hybrid offspring between the four species are commonly produced in studies [e.g. 86, 87]. These *Nasonia* hybrids do, however, present an interesting possibility of combining hybridization with knockdown polyploidization, to study variation in dosage and dominance effects from

different variations on which species contributes the polyploidized genome. Given its well annotated genome and the possibility of incorporating eye markers to easily track polyploid and nonpolypoid individuals (as in the WPL), *N. vitripennis* is uniquely well positioned for lab studies on neoautopolyploid phenotypes and gene expression.

Three of the challenges that successful autopolyploid lineages must overcome in their polyploid hop are meiotic problems, cellular changes, and gene dosage issues [28], but animal experiments demonstrating how this can be achieved have been limited because polyploids typically die in the first generation. We have shown that in a single species, there can be contrasting polyploid phenotypes for male fitness, meiosis, and cell reduction, but also mechanisms that are more consistent, such as sex-linked gene dosage that is resistant to ploidy change. Additional insight becomes possible with more *N. vitripennis* neopolyploid lines, which can easily be induced through knockdown of *transformer* [40], *transformer-2* [42] and *wasp overruler of feminization* [41] to produce founder diploid males and assay their phenotypes. Surveying the full range of polyploid biology for the long-established WPL, the neopolyploid single gene knockdown lines, and possibly induced WGD lines (e.g. adapting the methods of [30]) would further refine what constitutes a polyploidization pathway best suited to bridge the polyploid hop. The ability to create various long-term polyploid lineages, in conjunction with many existing resources for advanced genetic study of *N. vitripennis*, e.g. fully sequenced genome and transcriptome [35, 88], RNAi for gene knockdowns [89], and CRISPR for gene knockouts [90], mean that *N. vitripennis* can be developed into a comprehensive experimental model for animal polyploidy, a need in polyploid research that has been repeatedly called-for [2, 18, 28, 58, 59, 69].

## Supporting information

**S1 Fig. Individuals used for crosses and assays.** Horizontal lines indicate a cross and vertical lines indicate descent. Gray-filled symbols represent individuals that were used in assays. The untreated HVRx strain was used to generate control individuals for each generation and to continue breeding in the injected line. This background is represented with dashed lines. The *transformer* knockdown line (tKDL) was founded with F0 females injected with ds *tra* RNA. tKDL polyploids individuals from the F2 (female) and F3 (male) generations were not used for assays but were used to continue the line. The Whiting polyploid line (WPL) was used to produce inbred individuals of a long-established polyploid background to compare against outbred tKDL counterparts and is indicated by dotted lines. Individuals that were used for qPCR analysis of reference housekeeping genes *Ak3* and *ef1α* are marked with an asterisk (*). (TIF)

**S2 Fig. The ploidy typing process.** Horizontal lines indicate a cross and vertical lines indicate descent. Gray-filled symbols indicate individuals that were used in assays. The HVRx stock population background is denoted by dashed lines. The tKDL background is represented by solid lines. Ploidy was known a priori for WPL and control (HVRx) individuals. Typing for ploidy took place for tKDL haploid and diploid males as they have no distinguishing morphological markers. Ploidy typing of F1, F3, and F5 males used a two-step daughter-typing and flow cytometry approach. Ploidy could be inferred for F2 and F4 females through the ploidy of their corresponding fathers from the previous generation. The daughter-typing step is based on the lesser fecundity of triploid females. In a pilot study, control diploid HVRx females were capable of producing 60–90 offspring on three *Calliphora* sp. hosts. In contrast, triploid females of the WPL typically produce only four offspring (this paper). Hence, male ploidy can be partially determined from the fecundity of their daughters. For ploidy typing of the males, each male was mated to a virgin diploid HVRx female from the stock population. If the male

was haploid, it produced diploid female offspring. If the male was diploid, it produced triploid female offspring. Three daughters of each male were hosted on three hosts each (to account for poor reproduction of random females), and the offspring allowed to develop under standard conditions. For each male, if at least two of the three daughters produced over 50 offspring each, they were scored as diploid females, and their father assigned corresponding haploid status. If all three daughters produced 0–50 offspring, this possibly reflected lower fecundity of triploid daughters. If the daughters of a male were scored as possible triploids, either the males themselves or one of their representative daughters were processed with flow cytometry. Flow cytometry samples were prepared by removing the head from the body, placing it in a 1.5 ml Eppendorf tube, and freezing at -20˚C. Heads were then stained with propidium iodide (PI) using the protocol of (1) with the modification of using a MACSQuant® Analyzer 10 (Wageningen University, Laboratory of Genetics and University Medical Center Groningen, Central Flow Cytometry Unit) using the manufacturer's settings for PI: Blue 488 nm laser, filter 655–730 nm, channel B3, and PerCP-Vio680 dye. Specimen ploidy was assigned based on match to reference specimens of known ploidy i.e. a haploid HVRx control male, a diploid HVRx control female, and a triploid WPL female (1, 2). If daughters were being examined to infer father ploidy, a diploid daughter indicated a haploid father and triploid daughter indicated a diploid father. Samples with an unclear signal (<10% of samples) were discarded from analyses. (TIF)

**S1 Table. Primers used for qPCR of genes *Ak3* and *ef1α*.**
(DOCX)

**S1 Data. qPCR data for 5x dilution.**
(XLSX)

**S2 Data. qPCR data for 50x dilution.**
(XLSX)

**S3 Data. Wing setae counts.**
(XLSX)

## Acknowledgments

We thank Anna Rensink, Elena Dalla Benetta, and Marloes van Leussen for help with RNAi and qPCR analyses. We are also grateful to Demi Engels, Sylvia Gerritsma, Yuan Zou, Sander Visser, Thanh Ta and Elzemiek Geuverink for help with data collection. Special thanks go to Xuan Li and Eveline C. Verhulst for their encouragement on the writing of this manuscript, and the reviewers for their advice to improve the text.

## Author Contributions

**Conceptualization:** Kelley Leung.

**Data curation:** Kelley Leung.

**Formal analysis:** Kelley Leung.

**Funding acquisition:** Louis van de Zande, Leo W. Beukeboom.

**Investigation:** Kelley Leung.

**Methodology:** Kelley Leung, Louis van de Zande.

**Project administration:** Louis van de Zande, Leo W. Beukeboom.

**Resources:** Leo W. Beukeboom.

**Supervision:** Louis van de Zande, Leo W. Beukeboom.

**Writing – original draft:** Kelley Leung.

**Writing – review & editing:** Kelley Leung, Louis van de Zande, Leo W. Beukeboom.

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
