## [Decision Letter · Decision Letter 0]

15 Dec 2022

PONE-D-22-23470Effects of polyploidization and their evolutionary implications are revealed by heritable polyploidy in the haplodiploid wasp Nasonia vitripennisPLOS ONE

Dear Dr. Leung,

Thank you for submitting your manuscript to PLOS ONE. After careful consideration, we feel that it has merit but does not fully meet PLOS ONE’s publication criteria as it currently stands. Therefore, we invite you to submit a revised version of the manuscript that addresses the points raised during the review process.

We look forward to receiving your revised manuscript.

Kind regards,

Kun Lu, Ph.D.

Academic Editor

PLOS ONE

Journal Requirements:

3. Please upload a new copy of Figure 2 as the detail is not clear. Please follow the link for more information: https://blogs.plos.org/plos/2019/06/looking-good-tips-for-creating-your-plos-figures-graphics/" https://blogs.plos.org/plos/2019/06/looking-good-tips-for-creating-your-plos-figures-graphics/

Reviewers' comments:

Reviewer's Responses to Questions

**Comments to the Author**

1. Is the manuscript technically sound, and do the data support the conclusions?

Reviewer #1: Yes

Reviewer #2: Yes

2. Has the statistical analysis been performed appropriately and rigorously? 

Reviewer #1: I Don't Know

Reviewer #2: Yes

3. Have the authors made all data underlying the findings in their manuscript fully available?

Reviewer #1: Yes

Reviewer #2: Yes

4. Is the manuscript presented in an intelligible fashion and written in standard English?

Reviewer #1: Yes

Reviewer #2: Yes

5. Review Comments to the Author

Reviewer #1: This paper describes some interesting differences between a spontaneous, long domesticated line of polyploid wasps, and wasps that have been made polyploid experimentally in the work. Most major comments on the work have to do with displaying and interpreting the data, which has been skipped in many parts in the manuscript.

Major comments:

I don't understand why the first set of data described at length in the paper is placed as a supplementary table (Table S1). As a reader, I would like to be easily refer to the important data as I am reading, rather than searching around for another file. In general I see no reason for any of the figures or tables to be supplementary in this manuscript. As far as I know there are no page or figure limitations in this journal, so why not keep the work whole?

In addition to this, large amounts of data are simply listed in the text, for example about the female fecundity and a significant portion of the female mate choice work. All of this should be summarized in graphs or tables in order to make the results comprehensible to the reader, the current organization significantly decreases the impact of the work.

In regard to the cell size/number work, again, quanitifications should be presented in the figure, the wing pictures give an impression, but the result would be enhanced by being able to immediately see the presumably unbiased quantification.

Further, while I am prepared to believe that a cell only produces a single seta, it is not clear to me that all wings cells produce them, as there are some regions that seem not to have setae, and presumably there are multiple cells in this region. I would also feel more confident if more than one area were sampled, it is know that there are domains that differ across the wing in terms of size and cell density, and I think the authors need to show that this phenomenon is consistent and universal within, and ideally across tissues.

For the section on Gene Expression, the lack of a figure to help interpret the data is again an issue to be corrected

Minor comments:

Elipsis not needed here:

77 studying heritable polyploidy in insects to the same level as other groups... However, because of

Reviewer #2: The proposed manuscript “Effects of polyploidization and their evolutionary implications are revealed by heritable polyploidy in the haplodiploid wasp Nasonia vitripennis” greatly contributes to the study of evolutionary biology, especially effects and consequences of polyploidization tested on haplodiploid rare insect system - Nasonia vitripennis. I have not found any fundamental issues in the proposed manuscript. The manuscript is technically sound, presented in an intelligible fashion and written in good-quality English. Impact of the study is well highlighted. Statistical analyses were correctly selected and used. I appreciate the wide range of experiments and methodology. I have only some minor suggestions mentioned below.

1) Gene expression analysis and qPCR were performed but authors did not describe how RNA was extracted? From which tissue/organs. I did not find RNA extraction in the M&M section.

2) Line 63: Authors highlighted parthenogenesis and hermaphroditism as reproductive strategies that reduce problems of offspring viability. But there are other strategies such as gynogenesis, hybridogenesis, or kleptogenesis. I suggest to include all reproductive strategies or to mention parthenogenesis as one of more existing strategies (for example to add “e.g.” before parthenogenesis).

3) Authors stated that fish and amphibians are used as interesting models for study of polyploidization (lines 62-63, line 392). One of intriguing fish representatives is the Carassius complex that includes individuals with different ploidy levels, modes of reproduction, and modes of sex determination. Also, gynogenesis mentioned in point 2 is typical for Carassius. Another widely used model is amphibian genus Xenopus, which represents wide range of ploidy levels from diploid to dodecaploid - this is the highest variability in ploidy levels in tetrapods. I think these two groups deserve to be included in the proposed manuscript. I suggest two current works very close to the manuscript topic. The first work is a review that describes ploidy, reproduction, and cytogenetics of Carassius genus (Knytl et al 2022). The second one is a study in which divergent subgenome evolutionary trajectories were revealed within the allotetraploid Xenopus genome (Knytl et al 2023). Can authors add both (or at least one) of these papers to the intro/discussion?

Knytl M, Forsythe A, Kalous L. A Fish of Multiple Faces, Which Show Us Enigmatic and Incredible Phenomena in Nature: Biology and Cytogenetics of the Genus Carassius. Int J Mol Sci. 2022 Jul 22;23(15):8095. doi: 10.3390/ijms23158095. PMID: 35897665.

Knytl M, Fornaini NR, Bergelová B, Gvoždík V, Černohorská H, Kubíčková S, Fokam EB, Evans BJ, Krylov V. Divergent subgenome evolution in the allotetraploid frog Xenopus calcaratus. Gene. 2023 Jan 30;851:146974. doi: 10.1016/j.gene.2022.146974. Epub 2022 Oct 27. PMID: 36309242.

4) Line 77: there are three dots behind the word “groups”.

5) I am not sure if citation style is correctly used within the manuscript. For instance, citations in lines 98-99 and many others - if the citation is situated within a sentence, shouldn't it be written in the form “author et al”? Line 98: correct citations should be: “Leung et al [34] and Cournault and Aron [35]”.

6) Line 98: “(see references in Leung et al [34] and Cournault and Aron [35])”. Authors referred to references in two mentioned papers? Why did not authors refer directly to references? Or delete “references in” which would be clear.

7) Line 159: delete space in front of comma.

8) Line 276: please explain “CSD species” after the first mention.

9) Lines 286-287: is the plural “are” correct? Shouldn't it be “The cause for disparities is currently unknown”?

10) Line 303: Dot is missing behind citations [6,60,63].

11) Line 364: “that” is doubled.

After overall consideration of the manuscript quality I suggest minor revision. After following the above recommendations, the manuscript can meet the requirements of the Plos One journal.

6. PLOS authors have the option to publish the peer review history of their article (what does this mean?). If published, this will include your full peer review and any attached files.

Reviewer #1: No

Reviewer #2: **Yes: **Martin Knytl

---

## [Author Response · Author response to Decision Letter 0]

24 May 2023

Kelley Leung, PhD

Groningen Institute of Evolutionary Life Sciences

University of Groningen

Post Office Box 11103

9700 CC, Groningen, The Netherlands

k.leung@rug.nl

23 May, 2023

Dear Dr. Kun Lu and reviewers,

On behalf of all authors, I thank you for the decision to invite our manuscript, PONE-D-22-23470, “Effects of polyploidization and their evolutionary implications are revealed by heritable polyploidy in the haplodiploid wasp Nasonia vitripennis” for revision. We are grateful to the reviewers for their comments on how to improve the article. Our corresponding point-by-point changes are as follows:

(Reviewer comments are in italics, our responses are in regular text)

Reviewer #1: This paper describes some interesting differences between a spontaneous, long domesticated line of polyploid wasps, and wasps that have been made polyploid experimentally in the work. Most major comments on the work have to do with displaying and interpreting the data, which has been skipped in many parts in the manuscript.

Major comments:

I don't understand why the first set of data described at length in the paper is placed as a supplementary table (Table S1). As a reader, I would like to be easily refer to the important data as I am reading, rather than searching around for another file. In general I see no reason for any of the figures or tables to be supplementary in this manuscript. As far as I know there are no page or figure limitations in this journal, so why not keep the work whole?

In addition to this, large amounts of data are simply listed in the text, for example about the female fecundity and a significant portion of the female mate choice work. All of this should be summarized in graphs or tables in order to make the results comprehensible to the reader, the current organization significantly decreases the impact of the work.

We thank reviewer 1 for this astute observation. As indicated in our cover letter to PlosONE, this manuscript was previously submitted to other journals with different formatting. We made use of PlosONE’s policy that initial submissions may be open format to expedite the submission process. We agree with reviewer 1 that the study is better presented with data in the main text rather than supplementary materials, and with less reliance on description in the main text. We reorganize the manuscript accordingly:

Data for male mate competitions is now in Table 1.

Table S1, likelihood of a female to choose a male over a competition partner, is now Table 2, and in-text references now refer to a Table 2 instead of a Table S1. Accordingly, Table S2 on primer sequences becomes Table S1.

Data for female fecundity has now been moved to Table 3.

Data for setae count and wing size have now been moved to Table 4.

Data presentation in the main text has now also been rewritten for reading clarity and to emphasize core results instead of number listing.

In regard to the cell size/number work, again, quantifications should be presented in the figure, the wing pictures give an impression, but the result would be enhanced by being able to immediately see the presumably unbiased quantification.

Further, while I am prepared to believe that a cell only produces a single seta, it is not clear to me that all wings cells produce them, as there are some regions that seem not to have setae, and presumably there are multiple cells in this region. I would also feel more confident if more than one area were sampled, it is know that there are domains that differ across the wing in terms of size and cell density…

We agree with Reviewer 1 that analyses of the wings would be stronger with direct presentation of quantifiable data. Also, although we intentionally chose to consistently sample the same subsection of each wing to circumvent the problem of lack of visible setae in some regions (these exist in the juvenile stage but are not scorable in the adult stage because they are lost during the hardening of the exoskeleton (scelerization); Loehlin et al., 2010). We also agree that data on whole wings in addition to subsections would give stronger support to conclusions. As data collection of the whole wing too work intensive to do for every specimen, we include whole wing setae count for N=5 randomly chosen individuals for each test group. The trend of whole wings matches that of the subsections, so we are confident that cell size and number change with ploidy level as indicated in the manuscript. To make this data more apparent, they have now been moved to a Table 4. 

The whole wing counting is reflected in the materials and methods as:

“For each group, N=5 randomly selected individuals were also scored for setae count across the whole right forewing using the count tool in Photoshop.” (now lines 233-235)

…and I think the authors need to show that this phenomenon is consistent and universal within, and ideally across tissues.

We also agree that it would be ideal to sample more tissues to investigate cell reduction in Nasonia polyploids. Unfortunately, this is complicated by correct choice of tissues, which may have multinucleated endopolyploidy cells that would appear as false positives for polyploidy in flow cytometry. This would complicate interpreting significant differences between non-polyploid and polyploid individuals. For example, nearly all hymenopterans have male endopolyploid thoracic tissues (Aron et al., 2005). It is also difficult to count cells in organs accurately, as with 3-D organs, due to incomplete marker penetrance, ability to disassociate individual cells etc. We are testing protocol to do this for brain cells to consistent and accurately reflect the individual’s ploidy, but this is beyond the scope of the current manuscript. We do however now reflect the reviewer 1’s point in the discussion:

“Ideally, more tissues should be sampled to investigate cell reduction in Nasonia polyploids. Unfortunately, this is complicated by some tissues having multinucleated endopolyploid cells; for example, nearly all hymenopterans have male endopolyploid thoracic tissues [50]. These cells may be larger, preventing accurate comparison between non-polyploid and polyploid individuals. Cell-counting accuracy of 3D organs is also difficult due to incomplete marker penetrance or disassociation of individual cells. A good candidate for developing protocols for further cell reduction investigation are the brain cells, which consistently and accurately reflect the individual’s ploidy [72]. A reference atlas for the Nasonia brain has recently been published [73], so it should be possible to i.e. use exact neuron count to more fully assess cell reduction in Nasonia polyploids.” (now lines 567-576).

For the section on Gene Expression, the lack of a figure to help interpret the data is again an issue to be corrected 

These figure on gene expression was included in the original submission of this manuscript to PlosONE as supplementary material. We correct this to account for the reviewer’s comment by moving it into the main manuscript as Figure 2, to more directly indicate how gene expression is patterned by sex rather than ploidy level.

Minor comments:

Elipsis not needed here:

77 studying heritable polyploidy in insects to the same level as other groups... However, because of

We have removed the ellipsis and used a period instead. 

Reviewer #2: The proposed manuscript “Effects of polyploidization and their evolutionary implications are revealed by heritable polyploidy in the haplodiploid wasp Nasonia vitripennis” greatly contributes to the study of evolutionary biology, especially effects and consequences of polyploidization tested on haplodiploid rare insect system - Nasonia vitripennis. I have not found any fundamental issues in the proposed manuscript. The manuscript is technically sound, presented in an intelligible fashion and written in good-quality English. Impact of the study is well highlighted. Statistical analyses were correctly selected and used. I appreciate the wide range of experiments and methodology. I have only some minor suggestions mentioned below.

1) Gene expression analysis and qPCR were performed but authors did not describe how RNA was extracted? From which tissue/organs. I did not find RNA extraction in the M&M section. 

The extraction was only briefly alluded to, by referencing the procotol of Dalla Benetta et al., 2019. We added more detail by amending the M&M section “Gene dosage of housekeeping genes Ak3 and ef1α” to:

“To assess the effect of ploidy and sex on gene expression, we quantified the absolute expression of two housekeeping genes Adenylate kinase 3 (Ak3) and elongation factor 1 alpha (ef1α) for all backgrounds (N=5 each). We followed the protocol of Dalla Benetta et al., [92] with the following modifications. In brief, RNA was extracted from heads and abdomens separately. Each wasp’s head was individually placed into a 1.5 ml Eppendorf tube. The same was done for abdomens. Thoraxes were discarded due to likely endopolyploidy [36]. Samples were immediately frozen at -80° C, extracted using TriZol (Invitrogen, Carlsbad, CA, USA), and cDNA conversion performed according to manufacturer’s instructions. Five technical replicates for each sample type were performed to control for pipetting error. qPCRs were run for Ak3 and ef1α (primers in Table S1) on the Applied Biosystems 7300 system. Reactions were first performed at a 5x dilution (Fig.2, S1 file) in the event of large amount of RNAi producing asymptotic readings instead of accurately measuring the absolute expression levels. Analyses were then also run for 50x dilutions to test for consistency (S2 file). The qPCR reactions were run with 3 min of activation phase at 95°C, followed by 35 cycles of: 15s at 95°C, 30 s at 56°C, and 30 s at 72°C. Note that in typical qPCR reactions these two genes are used as reference for RNA quantity, but in this assay we use the amplification amount as a direct measure for gene expression level as function of ploidy and sex.” (Now lines 237-254) 

2) Line 63: Authors highlighted parthenogenesis and hermaphroditism as reproductive strategies that reduce problems of offspring viability. But there are other strategies such as gynogenesis, hybridogenesis, or kleptogenesis. I suggest to include all reproductive strategies or to mention parthenogenesis as one of more existing strategies (for example to add “e.g.” before parthenogenesis).

We thank reviewer 2 for this this more comprehensive representation of reproductive strategies that facilitate offspring survival in neopolyploids. We take his second suggestion to add “e.g.,” before “parthenogenesis”, as we do not have specific references for hybridogenesis or kleptogenesis.

3) Authors stated that fish and amphibians are used as interesting models for study of polyploidization (lines 62-63, line 392). One of intriguing fish representatives is the Carassius complex that includes individuals with different ploidy levels, modes of reproduction, and modes of sex determination. Also, gynogenesis mentioned in point 2 is typical for Carassius. Another widely used model is amphibian genus Xenopus, which represents wide range of ploidy levels from diploid to dodecaploid - this is the highest variability in ploidy levels in tetrapods. I think these two groups deserve to be included in the proposed manuscript. I suggest two current works very close to the manuscript topic. The first work is a review that describes ploidy, reproduction, and cytogenetics of Carassius genus (Knytl et al 2022). The second one is a study in which divergent subgenome evolutionary trajectories were revealed within the allotetraploid Xenopus genome (Knytl et al 2023). Can authors add both (or at least one) of these papers to the intro/discussion?

Knytl M, Forsythe A, Kalous L. A Fish of Multiple Faces, Which Show Us Enigmatic and Incredible Phenomena in Nature: Biology and Cytogenetics of the Genus Carassius. Int J Mol Sci. 2022 Jul 22;23(15):8095. doi: 10.3390/ijms23158095. PMID: 35897665.

Knytl M, Fornaini NR, Bergelová B, Gvoždík V, Černohorská H, Kubíčková S, Fokam EB, Evans BJ, Krylov V. Divergent subgenome evolution in the allotetraploid frog Xenopus calcaratus. Gene. 2023 Jan 30;851:146974. doi: 10.1016/j.gene.2022.146974. Epub 2022 Oct 27. PMID: 36309242.

We agree with the reviewer that these two references are relevant to the manuscript, and include them in the discussion as: 

“The types of polyploid questions investigated in other animal systems are also perhaps illustrative of the specific advantages and limitations of neopolyploid research in Nasonia. For example, both the fish genus Carrasius and the frog lab model Xenopus also exhibit a range of ploidies. Carrasius specifically has several modes of sexual and asexual reproduction and varying degrees of fecundity, and work has been done to link these factors to ploidy [84]. Xenopus has the widest range of ploidy levels in tetrapods (diploid to dodecaploid), originating from species hybridization events (allopolyploidy). It has been used for example to study the evolution of subgenomes over evolutionary time [85]. Unlike Crassius and Xenopus, Nasonia polyploidy has not been recorded in nature, so it cannot be used to study real-world evolution of polyploid population dynamics (although artificial lab populations are potentially possible). Furthermore, there is no evidence that Nasonia allopolyploids are possible, although hybrid offspring between the four species are commonly produced in studies [e.g. 86,87]. These Nasonia hybrids do, however, present an interesting possibility of combining hybridization with knockdown polyploidization, to study variation in dosage and dominance effects from different variations on which species contributes the polyploidized genome. Given its well annotated genome and the possibility of incorporating eye markers to easily track polyploid and nonpolypoid individuals (as in the WPL), N. vitripennis is uniquely well positioned for lab studies on neoautopolyploid phenotypes and gene expression.” (now lines 640-658)

4) Line 77: there are three dots behind the word “groups”.

We have deleted the three dots after “groups” (now line 85)

5) I am not sure if citation style is correctly used within the manuscript. For instance, citations in lines 98-99 and many others - if the citation is situated within a sentence, shouldn't it be written in the form “author et al”? Line 98: correct citations should be: “Leung et al [34] and Cournault and Aron [35]”.

We have corrected all of these citations. 

6) Line 98: “(see references in Leung et al [34] and Cournault and Aron [35])”. Authors referred to references in two mentioned papers? Why did not authors refer directly to references? Or delete “references in” which would be clear.

We delete “references in” 

7) Line 159: delete space in front of comma.

We have deleted this comma 

8) Line 276: please explain “CSD species” after the first mention.

We explain “CSD species” with the additional text,

 “as in species with complementary sex determination (CSD)[31–33,60–62]. For CSD, hemizygotes for a csd locus or loci are haploid males and heterozygotes are diploid females, but homozygotes are typically reproductively impaired males. It is unclear what proportions of Hymenoptera are CSD versus non-CSD species.” (now lines 494-498)

9) Lines 286-287: is the plural “are” correct? Shouldn't it be “The cause for disparities is currently unknown”?

The reviewer is correct, we change the verb to “is”. 

10) Line 303: Dot is missing behind citations [6,60,63].

Dot was added after citations.

11) Line 364: “that” is doubled.

Deleted one “that” (now line xxx)

The editor has further required us to

We have followed these templates in the revised manuscript.

We have corrected this information.

And the ‘Funding information’ and ‘Financial Disclosure’ sections of the submission process.

3. Please upload a new copy of Figure 2 as the detail is not clear. Please follow the link for more information: https://blogs.plos.org/plos/2019/06/looking-good-tips-for-creating-your-plos-figures-graphics/" https://blogs.plos.org/plos/2019/06/looking-good-tips-for-creating-your-plos-figures-graphics/

We uploaded a new copy of Figure 2 (the gene expression figure) that is larger and has better resolution.

The co-authors also saw some opportunities to improve upon the original text and so made some minor voluntary edits throughout the text (e.g. changing unclear wording of sentences, deleting unnecessary excess information in introduction and discussion sections).

Once again, we are pleased with the overall positive response of the reviewers. We hope that these revisions are sufficient and would be glad to address any further points.

Sincerely,

Kelley Leung, Dr.

Leo W. Beukeboom, Prof. Dr.

Louis van de Zande, Dr.

---

## [Decision Letter · Decision Letter 1]

26 Jun 2023

Effects of polyploidization and their evolutionary implications are revealed by heritable polyploidy in the haplodiploid wasp Nasonia vitripennis

PONE-D-22-23470R1

Dear Dr. Kelley,

We’re pleased to inform you that your manuscript has been judged scientifically suitable for publication and will be formally accepted for publication once it meets all outstanding technical requirements.

Kind regards,

Kun Lu, Ph.D.

Academic Editor

PLOS ONE

Additional Editor Comments (optional):

Please carefully correct typing errors based on reviewer's comments.

Reviewers' comments:

Reviewer's Responses to Questions

**Comments to the Author**

1. If the authors have adequately addressed your comments raised in a previous round of review and you feel that this manuscript is now acceptable for publication, you may indicate that here to bypass the “Comments to the Author” section, enter your conflict of interest statement in the “Confidential to Editor” section, and submit your "Accept" recommendation.

Reviewer #1: All comments have been addressed

Reviewer #2: All comments have been addressed

2. Is the manuscript technically sound, and do the data support the conclusions?

Reviewer #1: Yes

Reviewer #2: Yes

3. Has the statistical analysis been performed appropriately and rigorously? 

Reviewer #1: Yes

Reviewer #2: Yes

4. Have the authors made all data underlying the findings in their manuscript fully available?

Reviewer #1: Yes

Reviewer #2: Yes

5. Is the manuscript presented in an intelligible fashion and written in standard English?

Reviewer #1: Yes

Reviewer #2: Yes

6. Review Comments to the Author

Reviewer #1: The authors have adequately addressed the reviewer comments, and the manuscript is much improved, and is now suitable for publication.

Reviewer #2: It has been quite a long time since I reviewed the first version of the manuscript “Effects of polyploidization and their evolutionary implications are revealed by heritable polyploidy in the haplodiploid wasp Nasonia vitripennis”. I think the manuscript is presented on the high level of scientific soundness, originality, and English grammatical accuracy. I also believe that it will be of interest to a readership focused on polyploidy research, and therefore will also have a good citation ratio. The authors followed all my recommendations and I thank them for their cooperation. I only recommend reading the whole manuscript several times and fixing “cosmetic” errors and typos (italics, citations style…). For example, line 591: modify “Crassius and Xenopus “to Carassius and Xenopus” (insert “a” to “Crassius” and use italics for “Xenopus”).

After overall consideration of the manuscript quality I suggest minor revision, but the manuscript already meets the requirements of the Plos One journal. Looking forward to seeing the published version.

7. PLOS authors have the option to publish the peer review history of their article (what does this mean?). If published, this will include your full peer review and any attached files.

Reviewer #1: No

Reviewer #2: **Yes: **Martin Knytl

---

## [Editor Report · Acceptance letter]

2 Jul 2023

PONE-D-22-23470R1 

Effects of polyploidization and their evolutionary implications are revealed by heritable polyploidy in the haplodiploid wasp *Nasonia vitripennis*

Dear Dr. Leung:

I'm pleased to inform you that your manuscript has been deemed suitable for publication in PLOS ONE. Congratulations! Your manuscript is now with our production department. 

Kind regards, 

on behalf of

Dr. Kun Lu 

Academic Editor

PLOS ONE